# The Protective Effects of Neoastilbin on Monosodium Urate Stimulated THP-1-Derived Macrophages and Gouty Arthritis in Mice through NF-κB and NLRP3 Inflammasome Pathways

**DOI:** 10.3390/molecules27113477

**Published:** 2022-05-28

**Authors:** Wenjing Xu, Fenfen Li, Xiaoxi Zhang, Chenxi Wu, Yan Wang, Yanjing Yao, Daozong Xia

**Affiliations:** 1School of Pharmaceutical Sciences, Zhejiang Chinese Medical University, Hangzhou 310053, China; xwj18770322597@163.com (W.X.); ncusklifenfen@163.com (F.L.); 202011014011089@zcmu.edu.cn (C.W.); 202011113911046@zcmu.edu.cn (Y.W.); mianmian117@163.com (Y.Y.); 2Academy of Chinese Medical Sciences, Zhejiang Chinese Medical University, Hangzhou 310053, China; 15867976778@163.com

**Keywords:** gouty arthritis, neoastilbin, NLRP3 inflammasome, NF-κB pathway

## Abstract

Gouty arthritis (GA) is a frequent inflammatory disease characterized by pain, swelling, and stiffness of joints. Neoastilbin is a flavonoid isolated from the rhizome of *Smilax glabra*, which possesses various anti-inflammatory effects. However, the mechanism of neoastilbin in treating GA has not yet been clarified. Thus, this study was to investigate the protective effects of neoastilbin in both monosodium urate (MSU) stimulated THP-1-derived macrophages and the animal model of GA by injecting MSU into the ankle joints of mice. The levels of key inflammatory cytokines in MSU stimulated THP-1-derived macrophages were detected by enzyme-linked immunosorbent assay (ELISA) kits. Protein expressions of nuclear factor kappa B (NF-κB) and NOD-like receptor protein 3 (NLRP3) inflammasome pathways were further detected by Western blotting. In addition, swelling degree of ankle joints, the levels of inflammatory factors, infiltration of inflammatory cells and the expressions of related proteins were determined. Swelling degree and histopathological injury in ankle joints of MSU-injected mice were significantly decreased after being treated with neoastilbin. Moreover, neoastilbin significantly diminished the secretion of interleukin-1β (IL-1β), interleukin-6 (IL-6) and tumor necrosis factor-α (TNF-α), suppressing the activation of NF-κB and NLRP3 inflammasome pathways in both MSU stimulated THP-1-derived macrophages and the mouse model of GA. In summary, neoastilbin could alleviate GA by inhibiting the NF-κB and NLRP3 inflammasome pathways, which provided some evidence for neoastilbin as a promising therapeutic agent for GA treatment.

## 1. Introduction

Gouty arthritis (GA) is a common metabolic disorder caused by the deposition of monosodium urate (MSU) crystals within articular cartilage, synovial sacs and other tissues [1]. MSU crystals deposited in the joints can activate the inflammasome, which further secretes pro-inflammatory cytokines such as interleukin-1β (IL-1β), interleukin-6 (IL-6) and tumor necrosis factor-α (TNF-α), leading to inflammation and tissue destruction [2,3].

Inflammasomes are cytoplasmic polyprotein complexes. The concept was first proposed by Martinon et al. in 2002 [4]. To date, four kinds of inflammasomes have been identified, including NOD-like receptor protein 1 (NLRP1), NOD-like receptor protein 3 (NLRP3), NOD-like receptor C4 (NLRC4) and absent in melanoma 2 (AIM2) [5,6,7]. As part of the innate immune system, they can respond to various exogenous and endogenous irritants and play a critical role in protecting the host from numerous types of damage [8]. Particular attention should be paid to the NLRP3 inflammasome, which is composed of the innate immune sensor NLRP3, the adaptor molecule ASC and downstream protease pro-caspase-1 [9]. It is believed that activation of the NLRP3 inflammasome occurs in two sequential steps, termed as priming and assembly, respectively [10]. A priming step is initiated by Toll-like receptors (TLRs), leading to the activation of nuclear factor kappa B (NF-κB)-mediated signaling [11]. In the activation phase, the assembly of the NLRP3 inflammasome happens under various stimuli such as extracellular ATP, mitochondrial DNA and pathogen-related components, which gives rise to the autoproteolysis of pro-caspase-1 and the release of active Caspase-1, further cleaving pro-IL-1β and pro- IL-18 into their mature forms to trigger inflammatory responses such as GA [12,13,14].

In recent years, the incidence rate of GA has been sharply rising. Nonsteroidal anti-inflammatory drugs (NSAIDs), corticosteroids, colchicine, and analgesics are commonly prescribed to relieve GA accompanied by redness and swelling [15,16,17]. However, these agents often present several serious side effects, including renal toxicity and gastrointestinal bleeding, which threaten people’s lives [18,19]. Therefore, exploiting safer and more effective compounds is urgently needed.

*Smilax glabra* Roxb., commonly known as red smilax, is dried rhizome of the Liliaceae family. In previous studies, we found that several flavonoids isolated from *S. glabra*, such as astilbin, neoastilbin, isoastilbin and neoisoastilbin, all showed significant anti-inflammatory effects [20]. At present, many researchers have focused on the anti-inflammatory effect of astilbin, the main component of *S. glabra*, but few have paid attention to the role of its stereoisomer, neoastilbin, in GA [21,22,23]. Therefore, the aim of this study was to investigate the protective effects of neoastilbin on MSU stimulated THP-1-derived macrophages and GA in mice. On this basis, the mechanism of neoastilbin on GA through the NF-κB and NLRP3 inflammasome pathways was further explored.

## 2. Results

### 2.1. Effect of Neoastilbin on Cell Viability in THP-1-Derived Macrophages

The cytotoxicity of neoastilbin against THP-1-derived macrophages was determined by Cell Counting Kit-8 (CCK-8) assay. The results showed that neoastilbin had no obvious cytotoxicity toward the proliferation of THP-1-derived macrophages in the concentration range of 10–640 μM (Figure 1A), which laid a foundation for investigating the effect of different concentrations of neoastilbin on MSU-stimulated THP-1-derived macrophages. As shown in Figure 1B, neoastilbin could significantly improve cell viability in a concentration-dependent manner at 5–20 μM (*p* < 0.01). Neoastilbin at concentrations between 20–80 μM increased cell viability by approximately 40% compared to the lipopolysaccharide (LPS) + MSU group. Therefore, 5, 10 and 20 μM were chosen as the concentrations of neoastilbin in follow-up experiments.

### 2.2. Effect of Neoastilbin on the Secretion of Inflammatory Cytokines in THP-1-Derived Macrophages

Several key pro-inflammatory cytokines, such as IL-1β, IL-6 and TNF-α, could be released extracellularly to induce inflammation after MSU stimulation. We further examined the effects of neoastilbin on the levels of these cytokines in cell culture supernatant and compared with the effect of colchicine. The results revealed that the secretion of inflammatory cytokines increased significantly by three-fold upon MSU treatment when compared with the control group (*p* < 0.01). In contrast, the contents of IL-1β, IL-6 and TNF-α clearly decreased after neoastilbin administration when compared with the LPS + MSU group, especially with the high doses of neoastilbin (Figure 2).

### 2.3. Effects of Neoastilbin on Protein Expression of NF-κB and NLRP3 Inflammasome Pathways in THP-1-Derived Macrophages

Firstly, we explored the potential effects of neoastilbin on the NF-κB pathway in THP-1-derived macrophages. As shown in Figure 3, the phosphorylated IKKα, p65 and IκBα protein levels in THP-1-derived macrophages stimulated with MSU increased markedly compared with the protein expressions in the control group (*p* < 0.01). However, macrophages treated with neoastilbin showed a reduction trend in the phosphorylation of IKKα, p65 and IκBα in a dose-dependent way.

Then, the expression of p-p65 in the nucleus was further tested. As shown in Figure 4, NF-κB nuclear translocation occurred after MSU stimulation, and NF-κB phosphorylation protein p-p65 was detected in the nucleus. In the neoastilbin group, the expression of p-p65 in nucleus was not obvious.

Additionally, the results in Figure 5 show that the expressions of NLRP3, Caspase-1 and ASC proteins in cells stimulated by MSU significantly increased, whereas treatment with neoastilbin could obviously reduce the elevation of NLRP3, Caspase-1 and ASC protein expressions in MSU-stimulated THP-1-derived macrophages. Collectively, these results not only validated the anti-inflammatory effect of neoastilbin, but also further elucidated its anti-inflammatory mechanism in vitro.

### 2.4. Effect of Neoastilbin on MSU-Induced Ankle Swelling in GA Mice

The mouse model of GA was established by intra-articular injection of MSU to explore the effect of neoastilbin on GA. To assess the degree of swelling, the ankle swelling index of all mice was calculated by a toe volume measuring device. Significant increases in ankle swelling were observed in mice injected with intra-articular MSU compared to the control group, reaching a maximum at 6 h (Table 1). As shown in Figure 6, the ankle swelling in mice was significantly attenuated by neoastilbin at 25 mg/kg or 50 mg/kg compared to mice in the MSU group (*p* < 0.01), indicating neoastilbin could alleviate MSU-induced GA.

### 2.5. Effect of Neoastilbin on the Levels of Inflammatory Cytokines in GA Mice

In the occurrence and development of GA, IL-1β, IL-6 and TNF-α are well-established pro-inflammatory cytokines. The levels of IL-1β, IL-6 and TNF-α in mice ankle joints were detected by using ELISA kits to investigate the anti-inflammatory effect of neoastilbin on GA. As shown in Figure 7, the levels of IL-1β, IL-6 and TNF-α were significantly increased in MSU-injected mice compared with the control group, while obvious reductions in these inflammatory factors were seen in the ankle joints of mice treated with high-dose neoastilbin (*p* < 0.05). This phenomenon was also observed in the colchicine group.

### 2.6. Effect of Neoastilbin on MSU-Induced Inflammatory Infiltration in Ankle Joints

The role of neoastilbin in MSU-induced inflammation in vivo was then investigated. Histological analysis showed that the ankle joints of mice in the control group were intact without obvious inflammatory cell infiltration (Figure 8A). Conversely, MSU significantly increased the infiltration of inflammatory cells into the ankle joints, while the MSU group was also accompanied by injury and incompleteness of the ankle joint (Figure 8B). A noticeable decrease in inflammatory cell infiltration was observed in the colchicine-treated group (Figure 8C). Similarly, the neoastilbin treatment dose-dependently reduced infiltration of inflammatory cells and diminished the ankle joint damage due to GA (Figure 8D,E).

### 2.7. Effects of Neoastilbin on Protein Expression of NF-κB and NLRP3 Inflammasome Pathways in GA mice

We then investigated whether neoastilbin had an in vivo inhibitory effect on the activation of NF-κB and NLRP3 inflammasome pathways; the key targets in NF-κB and NLRP3 inflammasome pathways were determined by Western blotting. As shown in Figure 9, Figure 10 and Figure 11, marked increases in the protein expression levels of NLRP3, Caspase-1, ASC, p-IKKα, p-p65 and p-IκBα, as well as increased nuclear translocation of p65, were observed in ankle joints of mice in the MSU-induced group in comparison with the control group. In contrast, neoastilbin dose-dependently reduced the expression levels of these proteins, suggesting that inhibition of NF-κB and NLRP3 inflammasome pathways by neoastilbin could contribute to its anti-inflammatory effect in GA.

## 3. Discussion

GA is a recurrent immune inflammatory disease caused by disturbance of purine metabolism in the body. Due to the changing dietary structure and the rising rate of metabolic-associated diseases, the incidence of GA in the world has also increased correspondingly [24,25,26,27]. Although currently used drugs such as NSAIDs, corticosteroids and colchicine could effectively reduce inflammation and pain, their side effects could not be ignored [28,29]. Compared with these drugs, traditional Chinese medicine showed great potential and advantages with multiple targets and few side effects for GA. Flavonoids can be found in various plants and vegetables with structural diversity. As one of the secondary metabolites in plants, flavonoids possess important pharmacological activities such as antioxidant and anti-inflammatory effects. Neoastilbin is a flavonoid isolated from the rhizome of *S. glabra*. Cytotoxicity tests showed that neoastilbin did not inhibit cell proliferation at the concentration range of 10–640 μM, suggesting that it was a safe agent without obvious toxic or side effects (Figure 1A).

There are a large number of inflammatory factors such as IL-1β, IL-6 and TNF-α in the synovial fluid and serum of GA patients. IL-1β can promote the infiltration of immune cells and the recruitment of leukocytes. IL-6 is a vital mediator of cartilage degeneration, inflammatory cell aggregation and inflammatory reaction sustainability. TNF-α is mainly produced by activated macrophages in GA patients, and is a key factor leading to inflammatory reactions and joint destruction [30]. The inflammatory factors in mice ankle joints and THP-1-derived macrophage supernatants were detected by ELISA kits. The results showed that neoastilbin could reduce the secretion of IL-1β, IL-6 and TNF-α. Pathological sections also showed that there were fewer inflammatory cells infiltrating into joints compared with the MSU group, suggesting that neoastilbin could reverse the joint injury and joint incompleteness caused by MSU to a great extent. In addition, it was observed that the swelling of the ankle joints in mice in the neoastilbin group was obviously improved compared with that of mice in the MSU group. These findings preliminarily indicated the protective effect of neoastilbin on GA both in vivo and in vitro, which was consistent with previous reports that flavonoids had anti-inflammatory effects [31,32].

NF-κB is produced by homologous or heterodimerization of Rel family proteins, mainly in the form of p50 and p65 subunits. Under normal circumstances, NF-κB in the cytoplasm remains inactive and binds to the inhibitory protein IκB. Once stimulated, IκB kinase (IKK) participates in the phosphorylation of IκBα and then mediates the phosphorylation of NF-κB p65 [33]. Phosphorylated p65 enters the nucleus to induce the expression of inflammatory response-related genes such as IL-1β, IL-6 and TNF-α, that is, the activation of NF-κB is responsible for the levels of pro-inflammatory cytokines. Studies showed that glycyrrhizin treatment suppressed IL-1β-induced NF-κB phosphorylation in a mouse model of osteoarthritis (OA), significantly reducing IL-6, prostaglandin E2 (PGE2), nitric oxide (NO) and TNF-α levels [34]. Additionally, luteolin decreased the levels of cytokines in mice serum and intestine by inhibiting the activation of NF-κB, relieving DSS-induced colitis [35]. In order to further explore whether the anti-inflammatory effect of neoastilbin is related to activation of the NF-κB pathway, a more in-depth study was conducted. Western blotting analysis revealed that neoastilbin attenuated the phosphorylation of IKKα, p65 and IκBα in a dose-dependent way compared with the MSU-stimulated group (Figure 3 and Figure 9). In particular, the NF-κB pathway could be activated via nuclear translocation of p65 into the nucleus [36]. As expected, the nuclear translocation of p-p65 was significantly reduced in the neoastilbin group compared with the MSU-stimulated group (Figure 4 and Figure 10), which indicated that neoastilbin could downregulate key targets of the NF-κB pathway, inhibit the activation of the NF-κB pathway and reduce the release of IL-1β to exert protective effects.

The NLRP3 inflammasome consists of three components: NLRP3, ASC and proteases-1 protein. Among them, the C-terminal leucine-rich repeats (LRR) domain of NLRP3 receptor protein is responsible for recognizing ligands, whereas the N-terminal pyrin domain (PYD) participates in recruiting ASC. ASC, a connector protein, is composed of PYD and CARD domains. Once activated, NLRP3 acts as a sensor molecule that connects with ASC through PYD–PYD interaction. Subsequently, the aggregated ASC can recruit and activate pro-caspase-1 via the interaction between CARD and CARD. Activated pro-caspase-1 further leads to the maturation and release of pro-inflammatory cytokines [9]. It was reported that epigallocatechin-3-gallate (EGCG) alleviated GA by blocking the activation of NLRP3 inflammasome induced by various stimuli and synthesis of mitochondrial DNA in macrophages [37]. Amentoflavone (AM) and robustaflavone, the main active components of *Selaginella moellendorffii*, inhibited the formation of ASC specks and the expression of NLRP3 protein, exerting positive effects on GA [38]. Moreover, a variety of studies reported that the maturation and release of IL-1β were related to the activation of the NLRP3 inflammasome, and activation of NLRP3 was an important pathway for MSU to contribute to inflammatory responses. Thus, we speculated whether the anti-inflammatory mechanism of neoastilbin was connected with the NLRP3 inflammasome [39,40]. In vitro and in vivo experiments showed that the expression of NLRP3, Caspase-1 and ASC proteins increased significantly after MSU stimulation, indicating that the NLRP3 inflammasome might be activated. Interestingly, different doses of neoastilbin could reduce the expression of these proteins to varying degrees, which demonstrated that neoastilbin may exert an anti-inflammatory effect by inhibiting the activation of NLRP3 inflammasome (Figure 5 and Figure 11). In general, the results of cultured cell system and animal models in vivo indicated that neoastilbin had therapeutic effects on GA by inhibiting the activation of the NF-κB and NLRP3 inflammasome pathways.

In the present study, neoastilbin inhibited GA, including inflammatory cytokine release and inflammatory cell infiltration into the lesion site, mediated by the suppression of NF-κB and NLRP3 inflammasome pathways in THP-1-derived macrophages and GA mice. Considering that the NF-κB and NLRP3 inflammasome pathways are related to the development of many inflammatory diseases, neoastilbin may have potential clinical application value in gout, autoinflammatory syndrome or other NLRP3-driven diseases [41,42,43,44].

## 4. Materials and Methods

### 4.1. Samples and Reagents

Phorbol 12-myristate 13-acetate (PMA), lipopolysaccharide (LPS), monosodium urate (MSU), and colchicine were purchased from Sigma-Aldrich (St. Louis, MO, USA). Neoastilbin (purity ≥ 98%) was purchased from Sichuan Victory Biological Technology Co., Ltd. (Chengdu, China). Fetal bovine serum (FBS) was purchased from Gibco (Scoresby, Australia). Penicillin–streptomycin solution was purchased from HyClone (Logan, UT, USA). Cell Counting Kit-8 (CCK-8) was purchased from Biosharp Life Sciences (Beijing, China). Enzyme-linked immunosorbent assay (ELISA) kits for interleukin-1β (IL-1β), interleukin-6 (IL-6) and tumor necrosis factor-α (TNF-α) were purchased from MEIMIAN (Shanghai, China). NLRP3, ASC, Caspase-1, NF-κB p-p65, NF-κB p65, p-IKKα, IKKα, p-IκBα, IκBα, β-actin and Histone H3 antibodies were purchased from Cell Signaling Technology (Boston, MA, USA).

### 4.2. Cell Culture

Human monocyte cell line THP-1 cells were obtained from Shanghai Cell Bank of Chinese Academy of Sciences (Shanghai, China). THP-1 cells were seeded in 96-well plates (5 × 10^4^ cells/mL, 100 μL/well) with RPMI 1640 medium containing 10% FBS, 1% penicillin-streptomycin solution and 50 ng/mL PMA for 48 h to obtain THP-1-derived macrophages for cell viability assay. In the same way, THP-1 cells were seeded in 6-well plates (5 × 10^5^ cells/mL, 2 mL/well) to induce macrophages for ELISA and Western blotting analysis.

### 4.3. Cell Viability Assay by CCK-8

Cell viability was determined by the CCK-8 assay according to the manufacturer’s protocol. THP-1-derived macrophages were seeded in 96-well plates (5 × 10^4^ cells/mL, 100 μL/well) with complete medium consisting of various concentrations of neoastilbin for 24 h incubation; the drug concentrations were 10, 20, 40, 80, 160, 320, and 640 μM, respectively. After that, the medium was changed to RPMI 1640 basal medium containing 10% CCK-8 solution for another 2 h, then the absorbance of each well was measured at 450 nm wavelength with an enzyme microplate reader.

Subsequently, THP-1-derived macrophages were stimulated with LPS for 24 h. Different concentrations of neoastilbin (5, 10, 20, 40 and 80 μM) were added after washing out LPS with phosphate-buffered saline (PBS). After being pre-protected with neoastilbin for 0.5 h, cells were further stimulated with MSU suspension (500 μM) for another 23.5 h. Cell viability was determined with CCK-8 assay as previously described.

### 4.4. Inflammatory Cytokine Determination in THP-1-Derived Macrophages by ELISA Kits

THP-1-derived macrophages were seeded in 6-well plates (5 × 10^5^ cells/mL, 2 mL/well) for ELISA. After cells were stimulated with MSU as previously described, the cultured cell supernatant of each well was collected to determine the levels of IL-1β, IL-6, and TNF-α inflammatory factors according to the instructions of ELISA kits.

### 4.5. Analysis of Protein Expression of NF-κB and NLRP3 Inflammasome Pathways in THP-1-Derived Macrophages

Interventions of drugs in THP-1-derived macrophages were the same as described above. Cells were collected and lysed with radioimmunoprecipitation (RIPA) lysis buffer containing protease inhibitor and phosphatase inhibitor to extract total protein. Nuclear protein was extracted using NE-PER™ Kit (Thermo Scientific, Waltham, MA, USA). The BCA method (Beyotime, Shanghai, China) was applied to measure the concentration of protein, and the protein was denatured by heating in a water bath for 10 min. Equivalent amounts of protein (20 μg) were electrophoresed and separated by 10% sodium dodecyl sulfate–polyacrylamide gel electrophoresis (SDS–PAGE) and transferred onto polyvinylidene fluoride (PVDF) membranes. The membranes were blocked with 5% nonfat milk in Tris-buffered saline with 0.05% Tween-20 (TBST) for 1 h, then incubated with the primary antibodies overnight at 4 °C. After washing the membranes, they were incubated with horseradish peroxidase (HRP)-coupled secondary antibody (dilution, 1:5000; CST) for 1 h at room temperature and examined using an enhanced chemiluminescence (ECL) reagent. The band intensity was quantified using ImageJ software, version 1.53 (NIH, Bethesda, MD, USA).

### 4.6. Animals

Seven-to-eight-week-old male C57BL/6 mice (22 ± 2 g) were purchased from Shanghai SLAC Laboratory Animals Co., Ltd. (Shanghai, China. Certificate number: SCXK 2017-0005). The mice were housed in the specific-pathogen-free Animal Experimental Research Center of Zhejiang University of Traditional Chinese Medicine. Animal experiments were approved by the Animal Care and Use Committee of Zhejiang Chinese Medical University (Hangzhou, China. Permission number: SYXK 2018-0012).

After 7 days of acclimatization, the animals were randomly divided into the following five groups (*n* = 15 in each group): control group, MSU group (50 mg/mL), colchicine group (1 mg/kg), low-dosage neoastilbin-treated group (25 mg/kg), and high-dosage neoastilbin-treated group (50 mg/kg). The mice in the colchicine group were given colchicine by gavage, mice in the neoastilbin groups were given different concentrations of neoastilbin, and those in the control group and MSU group were intragastrically administered normal saline for 7 consecutive days. On the 6th day, the GA mouse model was established by intra-articular injection of MSU. One hour after intragastric administration, 0.025 mL of MSU suspension (50 mg/mL) was injected into the ankle joint cavity with a sterile syringe along the 45° direction of the dorsal side of the ankle joint, while mice in the control group were injected with an equal volume of normal saline.

### 4.7. Measurement of Ankle Swelling in GA Mice

To unify the toe volume measurement standard, a horizontal line was drawn on each mouse 0.5 cm above the ankle joint with a non-fading marker before the injection. Then, the toe volume of each mouse was measured by a toe volume measuring device before and 2, 4, 6, 10, and 24 h after MSU injection, respectively. The ankle swelling index was calculated using the following formula:$$\mathrm{Ankle}\;\mathrm{swelling}\;\mathrm{index}\;(\%)=\frac{\mathrm{Vafter}\;\mathrm{injection}\;-\mathrm{Vbefore}\;\mathrm{injection}}{\mathrm{Vbefore}\;\mathrm{injection}}\times 100\%$$

### 4.8. Inflammatory Cytokine Determination in GA Mice by ELISA Kits

The fur and muscles near the ankle joints were removed, then the ankle joints were ground with liquid nitrogen and dissolved fully in PBS. The supernatant was collected after centrifugation at 12,000 rpm and 4 °C for 10 min. The corresponding ELISA kit was used to measure the content of three inflammatory factors in GA mice.

### 4.9. Analysis of MSU-Induced Inflammatory Infiltration in Ankle Joints

All mice were sacrificed at 3 h after the final administration. The ankle joints of the mice were removed and fixed in buffered 10% formalin for 24 h before decalcification in ethylenediaminetetraacetic acid (EDTA). Then, the ankle joints were embedded in paraffin and stained with hematoxylin and eosin (H&E) to further analyze the pathological changes in the ankle joints.

### 4.10. Analysis of Protein Expression of NF-κB and NLRP3 Inflammasome Pathways in GA Mice

As previously described, total protein was extracted by lysing the ankle joints with RIPA lysis buffer containing protease and phosphatase inhibitors to detect the expression of NF-κB pathway-related proteins and NLRP3 inflammasome component proteins. At the same time, nuclear protein was separated by using a NE-PER™ Kit for detection of nuclear translocation of p65. The subsequent steps were the same as those described for cell experiments.

### 4.11. Statistical Analysis

All data were analyzed by one-way analysis of variance (ANOVA) followed by Dunnett’s multiple comparison tests using the statistical software SPSS 24.0 (SPSS Inc., Chicago, IL, USA). Data were expressed as means ± SD, and *p* < 0.05 was considered significantly different.

## 5. Conclusions

This study showed that neoastilbin had anti-inflammatory activity, which was manifested in reducing ankle swelling and inflammatory cell infiltration in GA mice. Furthermore, neoastilbin could exert anti-gout effects by inhibiting the activation of the NF-κB and NLRP3 inflammasome pathways both in vivo and in vitro. Therefore, the study could provide a theoretical basis for the treatment of GA with neoastilbin, thus contributing to the development and application of natural medicinal resources.

## Figures and Tables

**Figure 1 molecules-27-03477-f001:**
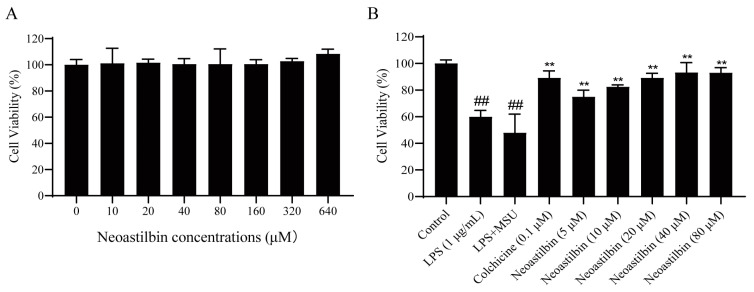
Effect of neoastilbin on cell viability in THP-1-derived macrophages. (**A**) THP-1-derived macrophages stimulated with various concentrations of neoastilbin. (**B**) The protective effect of neoastilbin for MSU-stimulated THP-1-derived macrophages. Data in the figures represent the means ± SD; significant differences among different groups are indicated as ## *p* < 0.01 vs. control group, ** *p* < 0.01 vs. LPS + MSU group (*n* = 3).

**Figure 2 molecules-27-03477-f002:**
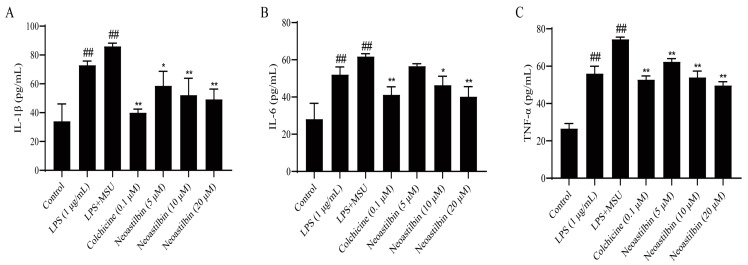
Effect of neoastilbin on the secretion of inflammatory cytokines in THP-1-derived macrophages. Macrophages were incubated for 24 h with LPS (1 µg/mL), followed by incubation for 0.5 h with neoastilbin, then incubating with MSU (500 µM) for an additional 23.5 h. Supernatants were analyzed by ELISA for IL-1β (**A**), IL-6 (**B**) and TNF-α (**C**) levels. Data in the figures represent the means ± SD; significant differences among different groups are indicated as ## *p* < 0.01 vs. control group, * *p* < 0.05, ** *p* < 0.01 vs. LPS+MSU group (*n* = 3).

**Figure 3 molecules-27-03477-f003:**
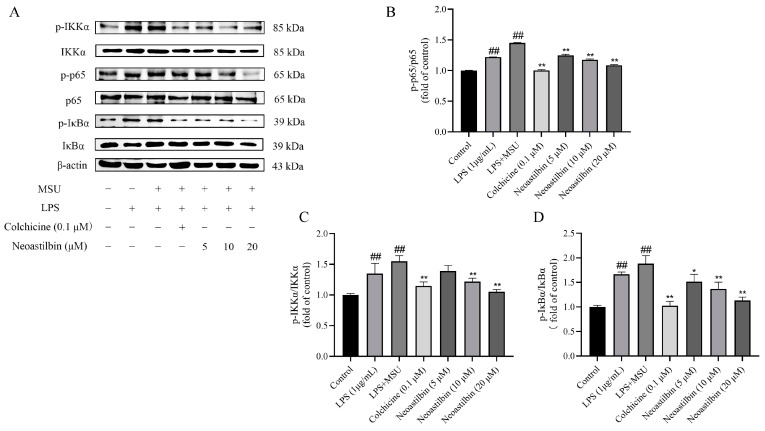
Effects of neoastilbin on protein expression of NF-κB pathway in THP-1-derived macrophages. (**A**) Phosphorylated and non-phosphorylated expressions of IKKα, p65 and IκBα. (**B**–**D**) The ratio of protein expression of p- IKKα/IKKα, p-p65/p65 and p- IκBα/IκBα. Data in the figures represent the means ± SD; significant differences among different groups are indicated as ## *p* < 0.01 vs. control group, * *p* < 0.05, ** *p* < 0.01 vs. LPS + MSU group (*n* = 3).

**Figure 4 molecules-27-03477-f004:**
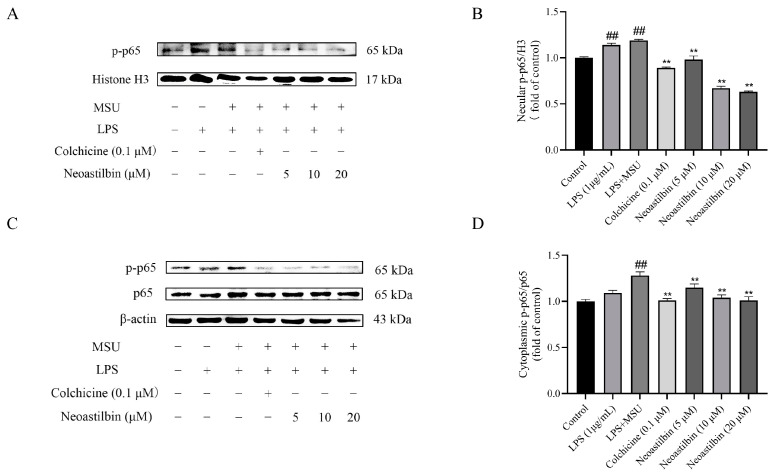
Effects of neoastilbin on protein expression of NF-κB p-p65 in THP-1-derived macrophages. (**A**) The protein expression of p-p65 in the nucleus. (**B**) The ratio of protein expression of p-p65/H3 in the nucleus. (**C**) Phosphorylated and non-phosphorylated expressions of p65 in the cytoplasm. (**D**) The ratio of protein expression of p-p65/p65 in the cytoplasm. Data in the figures represent the means ± SD; significant differences among different groups are indicated as ## *p* < 0.01 vs. control group, ** *p* < 0.01 vs. LPS + MSU group (*n* = 3).

**Figure 5 molecules-27-03477-f005:**
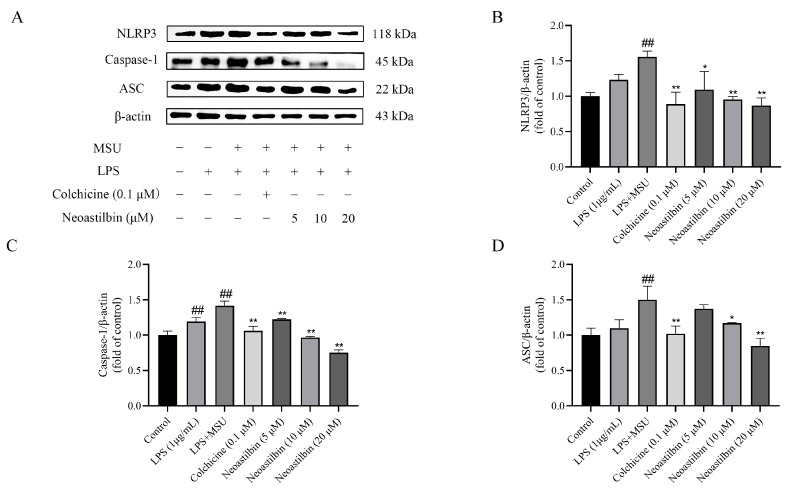
Effects of neoastilbin on protein expression of NLRP3 inflammasome in THP-1-derived macrophages. (**A**) The protein expressions of NLRP3, Caspase-1 and ASC. (**B**–**D**) The ratio of protein expression of NLRP3, Caspase-1 and ASC. Data in the figures represent the means ± SD; significant differences among different groups are indicated as ## *p* < 0.01 vs. control group, * *p* < 0.05, ** *p* < 0.01 vs. LPS + MSU group (*n* = 3).

**Figure 6 molecules-27-03477-f006:**
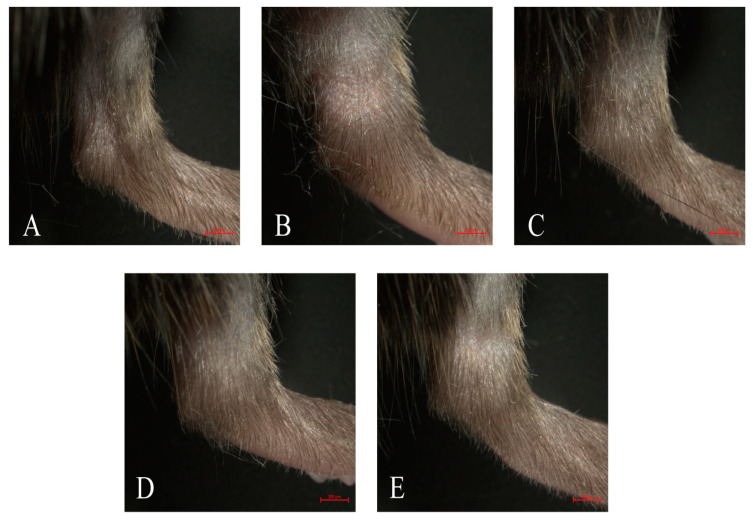
Effect of neoastilbin on MSU-induced ankle swelling in GA mice. Representative photographs of ankles 24 h after MSU injection: (**A**) control group, (**B**) MSU group, (**C**) colchicine group, (**D**) neoastilbin low-dose group, and (**E**) neoastilbin high-dose group.

**Figure 7 molecules-27-03477-f007:**
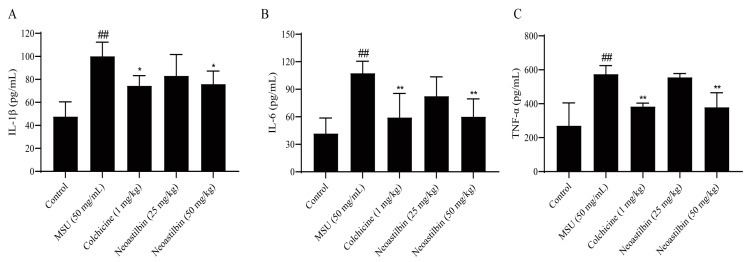
Effect of neoastilbin on the levels of inflammatory cytokines in GA mice. Mice were intragastrically administered neoastilbin (25 or 50 mg/kg) or colchicine (1 mg/kg) for 7 consecutive days. On the 6th day, MSU (50 mg/mL; 0.025 mL per mice) were injected into the ankle joints of the mice except the control group. After 24 h, ankle supernatants were analyzed by ELISA for IL-1β (**A**), IL-6 (**B**) and TNF-α (**C**) levels. Data in the figures represent the means ± SD; significant differences among different groups are indicated as ## *p* < 0.01 vs. control group, * *p* < 0.05, ** *p* < 0.01 vs. MSU group (*n* = 6).

**Figure 8 molecules-27-03477-f008:**
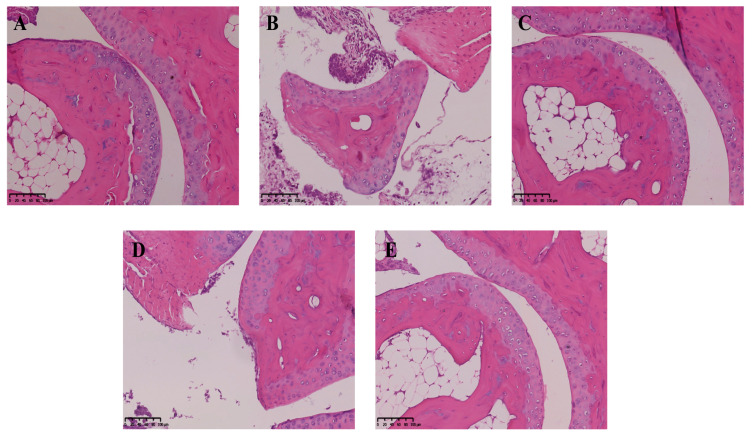
Effect of neoastilbin on MSU-induced inflammatory infiltration in ankle joints: (**A**) control group, (**B**) MSU group, (**C**) colchicine group, (**D**) neoastilbin low-dose group, and (**E**) neoastilbin high-dose group.

**Figure 9 molecules-27-03477-f009:**
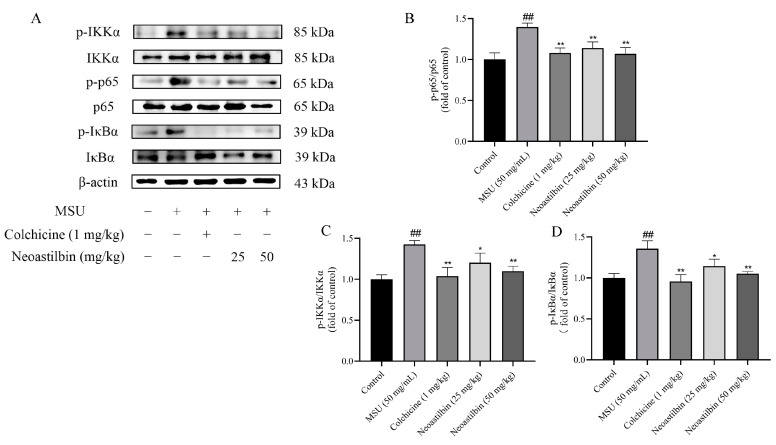
Effects of neoastilbin on protein expression of NF-κB pathway in GA mice. (**A**) Phosphorylated and non-phosphorylated expressions of IKKα, p65 and IκBα. (**B**–**D**) The ratio of protein expression of p- IKKα/IKKα, p-p65/p65 and p- IκBα/IκBα. Data in the figures represent the means ± SD; significant differences among different groups are indicated as ## *p* < 0.01 vs. control group, * *p* < 0.05, ** *p* < 0.01 vs. MSU group (*n* = 3).

**Figure 10 molecules-27-03477-f010:**
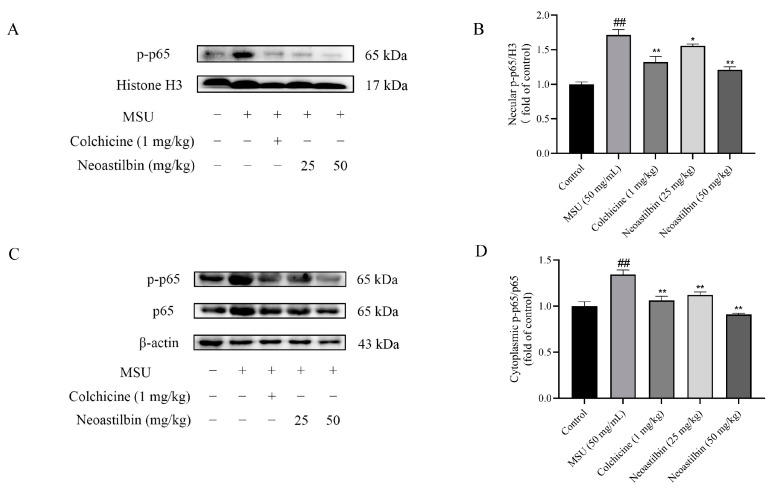
Effects of neoastilbin on protein expression of NF-κB p-p65 in GA mice. (**A**) The protein expression of p-p65 in the nucleus. (**B**) The ratio of protein expression of p-p65/H3 in the nucleus. (**C**) Phosphorylated and non-phosphorylated expressions of p65 in the cytoplasm. (**D**) The ratio of protein expression of p-p65/p65 in the cytoplasm. Data in the figures represent the means ± SD; significant differences among different groups are indicated as ## *p* < 0.01 vs. control group, * *p* < 0.05, ** *p* < 0.01 vs. MSU group (*n* = 3).

**Figure 11 molecules-27-03477-f011:**
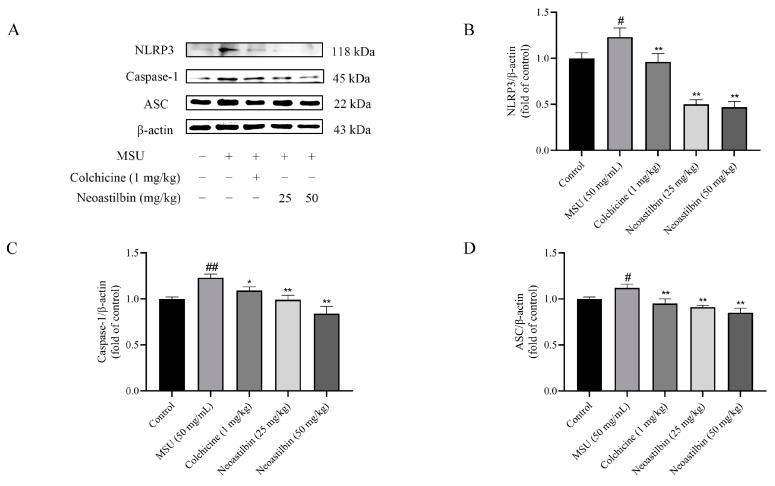
Effects of neoastilbin on protein expression of NLRP3 inflammasome in GA mice. (**A**) The protein expressions of NLRP3, Caspase-1 and ASC. (**B**–**D**) The ratio of protein expression of NLRP3, Caspase-1 and ASC. Data in the figures represent the means ± SD; significant differences among different groups are indicated as # *p* < 0.05, ## *p* < 0.01 vs. control group, * *p* < 0.05, ** *p* < 0.01 vs. MSU group (*n* = 3).

**Table 1 molecules-27-03477-t001:** Measurement of the ankle joint swelling in mice.

Group	Dosage/mg·kg^−1^	2 h	4 h	6 h	10 h	24 h
Control	-	20.35 ± 5.19	34.90 ± 4.75	32.89 ± 14.48	18.61 ± 9.61	15.25 ± 6.86
MSU	-	29.81 ± 9.14 #	50.05 ± 12.29 ##	66.84 ± 15.34 ##	55.93 ± 16.91 ##	54.26 ± 16.22 ##
MSU +Colchicine	1	32.25 ± 8.72	45.68 ± 7.62	49.17 ± 10.65 *	36.23 ± 13.21 *	20.64 ± 16.21 **
MSU +Neoastilbin	25	27.58 ± 11.54	49.75 ± 9.01	56.59 ± 6.77	50.44 ± 14.71	43.93 ± 8.17
	50	40.25 ± 10.08	47.58 ± 8.62	55.42 ± 10.43	38.43 ± 10.74 *	27.96 ± 7.49 **

Table values represent as means ± SD; significant differences among different groups are indicated as # *p* < 0.05, ## *p* < 0.01 vs. control group, * *p* < 0.05, ** *p* < 0.01 vs. MSU group (*n* = 10).

## Data Availability

All data generated or analyzed during the present study are included in this published article.

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
