# Peer review of "The Protective Effects of Neoastilbin on Monosodium Urate Stimulated THP-1-Derived Macrophages and Gouty Arthritis in Mice through NF-κB and NLRP3 Inflammasome Pathways"

_molecules, 2022, doi:10.3390/molecules27113477_

Round 1

Reviewer 1 Report

The current work investigated the anti-inflammatory effects of neoastilbine against MSU-induced gouty arthritis both in vitro and in vivo. The study is well-designed and well-represented. However, I have some suggestions before this manuscript can be accepted for publication. 

  1. It would be more appropriate to represent the data as mean ± SD rather than SEM.
  2. Lines 89, 160 and 167: You determined the (levels) not the (expression) of the investigated cytokines.
  3. The significance symbols should be unified in all figures. For example, ** should indicate significant difference vs  MSU group either at P<0.05 or P<0.01 in all figures (Please revise the figure legend of figure 1).
  4. Line 94: "which was comparable to colchicine". This could only be reported based on statistical comparison of the colchicine treated group with the groups treated with neoastilbin indicating which was/wasn't significantly different. The same applies to a similar note in line 164.
  5. The symbol # wasn't identified in figure 11 legend.
  6. The discussion should include a more detailed explanation of the NF-kB and the NLRP3 inflammasome pathways and the contributing role of each of the component biological markers that were detected in this study.
  7. There are several typos in the manuscript. Moreover, several abbreviations were used without being defined. Please revise carefully.

Author Response

Reply to the reviewer 1's questions:

Dear reviewer,

Thank you very much for your comments and suggestions. We tried our best to improve the manuscript and made all changes marked with red fonts in revised manuscript which will not influence the content and framework of the paper. We appreciate for your warm work, and hope the correction will meet with approval.

Question 1. It would be more appropriate to represent the data as mean ± SD rather than SEM.

Response: Thank you very much for your valuable suggestion. We have made corresponding modifications and marked with red fonts in the paper. As to Table 1, we have used mean ± SD represent the data in the original manuscript. However, we have made a mistake in the note of Table 1 as mean ± SEM. We have checked the data of Table 1 carefully, and the revisions have marked with red fonts in the revised manuscript. Please check them.

Question 2. Lines 89, 160 and 167: You determined the (levels) not the (expression) of the investigated cytokines.

Response: Thank you very much for your good advice. We have replaced "expression" with "levels" of the investigated cytokines as appropriate. Please see Lines 18, 92, 157, 161, 167, 253 and 256.

Question 3. The significance symbols should be unified in all figures. For example, ** should indicate significant difference vs MSU group either at P<0.05 or P<0.01 in all figures (Please revise the figure legend of figure 1).

Response: We are grateful to the reviewer for noticing this mistake in the figure legend of Figure 1. We are sorry for our oversight and this part has been modified. The significance symbols have been unified in all figures. Please check them.

Question 4. Line 94: "which was comparable to colchicine". This could only be reported based on statistical comparison of the colchicine treated group with the groups treated with neoastilbin indicating which was/wasn't significantly different. The same applies to a similar note in line 164.

Response: Special thanks to you for your good comments. We are sorry for our mistakes and have deleted the relevant words to avoid misunderstanding. Please see Lines 97 and 165.

Question 5. The symbol # wasn't identified in figure 11 legend.

Response: Thank you for your kind reminder. We have supplemented it in Figure 11 legend of our manuscript. Please see Line 216.

Question 6. The discussion should include a more detailed explanation of the NF-κB and the NLRP3 inflammasome pathways and the contributing role of each of the component biological markers that were detected in this study.

Response: We are deeply thankful for the critiques that have greatly improved our manuscript. In the discussion part, we have reviewed the relevant literature and made revisions. And the modified sections were marked with red fonts in the paper. Please check them.

Question 7. There are several typos in the manuscript. Moreover, several abbreviations were used without being defined. Please revise carefully.

Response: Thank you very much for your reminder. We apologize for the mistakes in this manuscript and the inconvenience they caused in your reading. We have carefully revised the manuscript for a long time and also invited native English speakers for language improvement. Moreover, the full descriptions of the abbreviations such as IL-1β, IL-6, TNF-α have been supplemented in the revised manuscript. Please check them.

Special thanks to you for your good comments and suggestions!

Reviewer 2 Report

This is a very interesting study with the objective of investigating the protective effects of neoastilbin in an animal model of gouty arthritis induced by monosodium urate (MSU). Moreover, the mechanism involved in the anti-inflammatory effect of this flavonoid was evaluated employing THP-1-derived macrophages stimulated by MSU. The study was carried out with the appropriate methodology. The results represent an important contribution to the literature by demonstrating that neoastilbin inhibits NF-kB and NLRP3 inflammasome pathways both in vivo and in vitro, and show that this flavonoid may be a promising therapeutic agent for gouty arthritis. 

The use of macrophages derived from the human monocyte lineage THP-1 to study the inhibitory effects of neostilbin on the NLRP3 inflammasome was important to show the mechanism involved in the anti-inflammatory activity of this flavonoid, in the two sequential steps, priming, and assembly of this inflammasome. The experimental model of gouty arthritis induced by MSU was very well explored and confirms the anti-inflammatory effect of neoastilbin by reducing the secretion of inflammatory cytokines and the inflammatory cells infiltrate in joints.

my few suggestions are related to the methodology.
In item 4.2. of material and methods, the concentration of THP-1 cells was not informed. Only in item 4.3. there is the information that THP-1-derived macrophages were seeded in 96-well plates at a density of 5 x 103 cells per well.
item 4.5. What was the protein concentration used in the Western blotting technique to analyze the total and nuclear protein of NF-kB and NLRP3 inflammasome?
item 4.7. Measurement of ankle swelling in GA mice: What is the meaning of the letter V in the formula of ankle swelling index?

Author Response

Reply to the reviewer 2's questions:

Dear reviewer,

Thank you very much for your comments and suggestions. We tried our best to improve the manuscript and made all changes marked with red fonts in revised manuscript which will not influence the content and framework of the paper. We appreciate for your warm work, and hope the correction will meet with approval.

In item 4.2. of material and methods, the concentration of THP-1 cells was not informed. Only in item 4.3. there is the information that THP-1-derived macrophages were seeded in 96-well plates at a density of 5×103 cells per well.

Response: Thank you very much for your comments. We have checked our original data and supplemented detail description of the concentration of THP-1 cells as comprehensively as possible. Please see item 4.2., 4.3. and 4.4 in the revised manuscript.

In item 4.5. What was the protein concentration used in the Western blotting technique to analyze the total and nuclear protein of NF-κB and NLRP3 inflammasome?

Response: Thank you very much for your kind reminder. The amounts of protein used in the Western blotting technique to analyze the total and nuclear protein of NF-κB and NLRP3 inflammasome were both 20 μg. We have supplemented it in the manuscript. Please see item 4.5 in the revised manuscript.

In item 4.7. Measurement of ankle swelling in GA mice: What is the meaning of the letter V in the formula of ankle swelling index?

Response: Thank you very much for your valuable proposal. We are sorry that this description has caused you confusion. The letter V in the formula of ankle swelling index represented the toe volume of mice. We have added detailed description in item 4.7. according to your constructive suggestions, and sincerely hope to receive better readability.

Special thanks to you for your good comments and suggestions!

Round 2

Reviewer 1 Report

The manuscript has been improved and can be accepted in the current form.